# The species composition—ecosystem function relationship: A global meta-analysis using data from intact and recovering ecosystems

Peter J. Carrick[1‡]*, Katherine J. Forsythe[2‡]

1 Plant Conservation Unit, Department of Biological Sciences, University of Cape Town, Rondebosch, South Africa, 2 Percy FitzPatrick Institute of African Ornithology, Department of Biological Sciences, University of Cape Town, Rondebosch, South Africa

‡ These authors are joint senior authors on this work.
* peter.carrick@uct.ac.za

**Data Availability Statement:** All relevant data are within the paper and its Supporting Information files.

## Abstract

The idea that biodiversity is necessary in order for ecosystems to function properly has long been used as a basic argument for the conservation of species, and has led to an abundance of research exploring the relationships between species richness and ecosystem function. Here we present a meta-analysis of global ecosystems using the Bray-Curtis index to explore more complex changes in the species composition of natural ecosystems, and their relationship with ecosystem functions. By using data recorded, firstly in reference sites and secondly in recovering sites, captured in restoration ecology studies, we pose the following questions: Firstly, how much variation is there in species composition and in ecosystem function in an intact ecosystem? Secondly, once an ecosystem has become degraded, is there a general relationship between its recovery in species composition and its recovery in ecosystem function? Thirdly, is this relationship the same for all types of ecosystem functions? Data from 21 studies yielded 478 comparisons of mean values for ecosystems. On Average, sites within the same intact natural ecosystems shared only a 48% similarity in species composition but were 69% similar in ecosystem functioning. In recovering ecosystems the relationship between species composition and ecosystem function was weak and saturating (directly accounting for only 2% of the variation). Only two of the six types of ecosystem function examined, biomass and biotic structure, showed a significant relationship with species composition, and the three types that measured soil functions showed no significant relationship. To date, most biodiversity—ecosystem function (BEF) research has been conducted in simplified ecosystems using the simple species richness metric. This study encourages a broader examination of the drivers of ecosystem functions under realistic scenarios of biodiversity change, and highlights the need to properly account for the extensive natural variation.

**Funding:** The authors received no specific funding for this work.

**Competing interests:** The authors have declared that no competing interests exist.

## Introduction

Concern over the effect of rapid biodiversity loss on an ecosystem's ability to function, and in turn, on its ability to provide humans with valuable ecosystem services, gave rise to the extensive field of research on biodiversity and ecosystem functioning (BEF). Since the 1980s, considerable research, both experimental and correlational, has focused on determining whether the functioning of an ecosystem is influenced by the number of species within it (for reviews see e.g. [1–9]).

In the last few years, consolidation of the BEF field, has revealed wide support for the following generalities:

1. An overall positive relationship exists between species richness and individual ecosystem functions [4,5].

2. The positive relationship between species richness and ecosystem functions is overwhelmingly non-linear and saturating [1,7,8,10], but neither the direction or the shape of the relationship is consistent within or across all studies [1,8].

3. The functional traits of species, or particular combinations of species, are often important in determining species interactions (including species interactions between trophic levels) and ecosystem functioning, rather than number of species alone [3,6,7,11–15].

4. The impact of species richness on ecosystem multifunctionality (the integration of the impact on a number of functions) is greater than on an average single ecosystem function [10,16,17].

5. Increasing species richness confers stability to ecosystem functions, i.e. the variance of an average ecosystem function is decreased in measurements across time or space as biodiversity increases [3,4,7,11; but see 1].

Increasingly, studies support the notion that the types of species in the ecosystem have a greater impact on individual ecosystem functions than simply the number of species in the ecosystem [3,10,18,19]. In fact, species can have positive, negative or neutral impacts on ecosystem function, depending on the type of ecosystem function and the species involved [15,20]. Real-world studies have also shown that the abundance of individuals (or the simply the abundance common species) can be the most important driver of ecosystem function [21]. Thus, there is a critical need to explore relationships driven by species composition rather than simply by species richness.

In this paper we pose allied but fundamentally different questions to traditional BEF studies which have focussed on the relationship between the number of species in the ecosystem (i.e. species richness, or often more precisely, species density; [22]) and various measures of ecosystem function. Here, our focus is on the relationship between the species composition of an ecosystem (i.e. community composition or community assemblage) and ecosystem function.

Firstly, within intact natural ecosystems, what is the range of variation within species composition and within ecosystem function, and do the two differ? Secondly, among natural ecosystems that have become degraded by some means, does a general relationship exist between recovery in species composition and recovery in ecosystem function, and what is the strength and shape of this relationship? Thirdly, does the relationship between recovery in species composition and recovery in ecosystem function differ among types of functions and types of ecosystems?

We explore this relationship by conducting a meta-analysis using data from the rapidly expanding field of restoration ecology. A number of researchers have emphasised the need for future BEF research to work on more realistic scenarios, where human activities are modifying biodiversity, and to use more complex, real-world ecosystems already undergoing compositional shifts [7,23,24].

Considerable ecological research has been focussed on the restoration of natural ecosystems, and the data from these studies provide ideal opportunities for addressing large-scale BEF questions [15,25–27]. Ecological restoration studies are useful for such research because they increasingly use multiple reference sites [28, e.g. 29,30], which allows quantification of the inherent heterogeneity and the variety of states in which intact ecosystems naturally occur, both spatially and temporally [31,32]. Restoration studies tend to measure a broader range of ecological functions than most BEF studies [see 28], making them more representative of the range of functions that occur in natural ecosystems. Various measures of species identity and profusion (abundance, cover, biomass etc.) from restoration studies can usefully be integrated with similarity indices (e.g. the Bray-Curtis similarity index) for comparing the species composition of ecosystems. These indices render a more complete view of the ecosystems' biota than species richness alone. Finally, restoration studies allow comparisons between species composition and ecosystem function to be made across the full spectrum of ecosystem conditions, from completely degraded, through recovering, to intact natural ecosystems [e.g. 33] and thus better reflect many of the states and complexities of the world's ecosystems.

Data from restoration sites provide snapshots at arbitrary points in time, to meaningfully quantify the difference of a recovering site from their intact ecosystems in terms of species composition and ecosystem function. We were not interested in whether sites were moving on any trajectory towards degradation or towards recovery. Rather, species composition was compared with ecosystem function at whatever point it was measured in a restoration study, the pattern from all the available points applied to a best-fit relationship and the robustness of this relationship tested. To our knowledge this is the first global study to explicitly explore the relationship between species composition and ecosystem function across ecosystem and function types, and to do so using real-world ecosystems.

## Materials and methods

### Literature search

A literature search was conducted in Web of Knowledge (Thomson Reuters Web of Knowledge) using the terms (RESTOR* OR REHABILIT* OR REFOREST*) AND (ECOLOG* OR ECOSYSTEM OR ENVIRON*) AND (FUNCTION* OR PROCESS* OR SERVICE*) AND (COMPOSITION OR BIODIVERSITY OR DIVERSITY). The resulting studies were filtered by first examining titles, then abstracts for broad relevance, and finally the selected studies were read in full [34]. From these studies we selected only those that met the following two criteria: studies that measured species-level data at restoration sites, together with at least one measure of ecosystem function (in order to quantify both species composition and ecosystem function at the same point on the trajectory of restoration in each ecosystem); additionally species composition and ecosystem functions must also have been recorded in multiple reference sites (in order to quantify the range of natural or intact conditions for each ecosystem). Authors of these studies were then contacted for the raw data: abundance, cover or biomass of each individual species, and metrics for all measured ecosystem functions within each site.

The literature search produced 4072 studies. After examining these studies for title relevance, abstracts and then in full, 67 studies meeting our criteria remained and requests for the data were then made to the authors. Studies were excluded at this last stage because the data provided were not sufficient as they failed to either adequately measure species composition or to provide suitable reference sites. The authors of twenty studies were able to provide suitable raw data, and an additional five studies were included as sufficient data was provided within the publications themselves, or appendices and supplementary material (Fig 1). Twenty-one studies reported on the species composition of plants and four on the species composition of

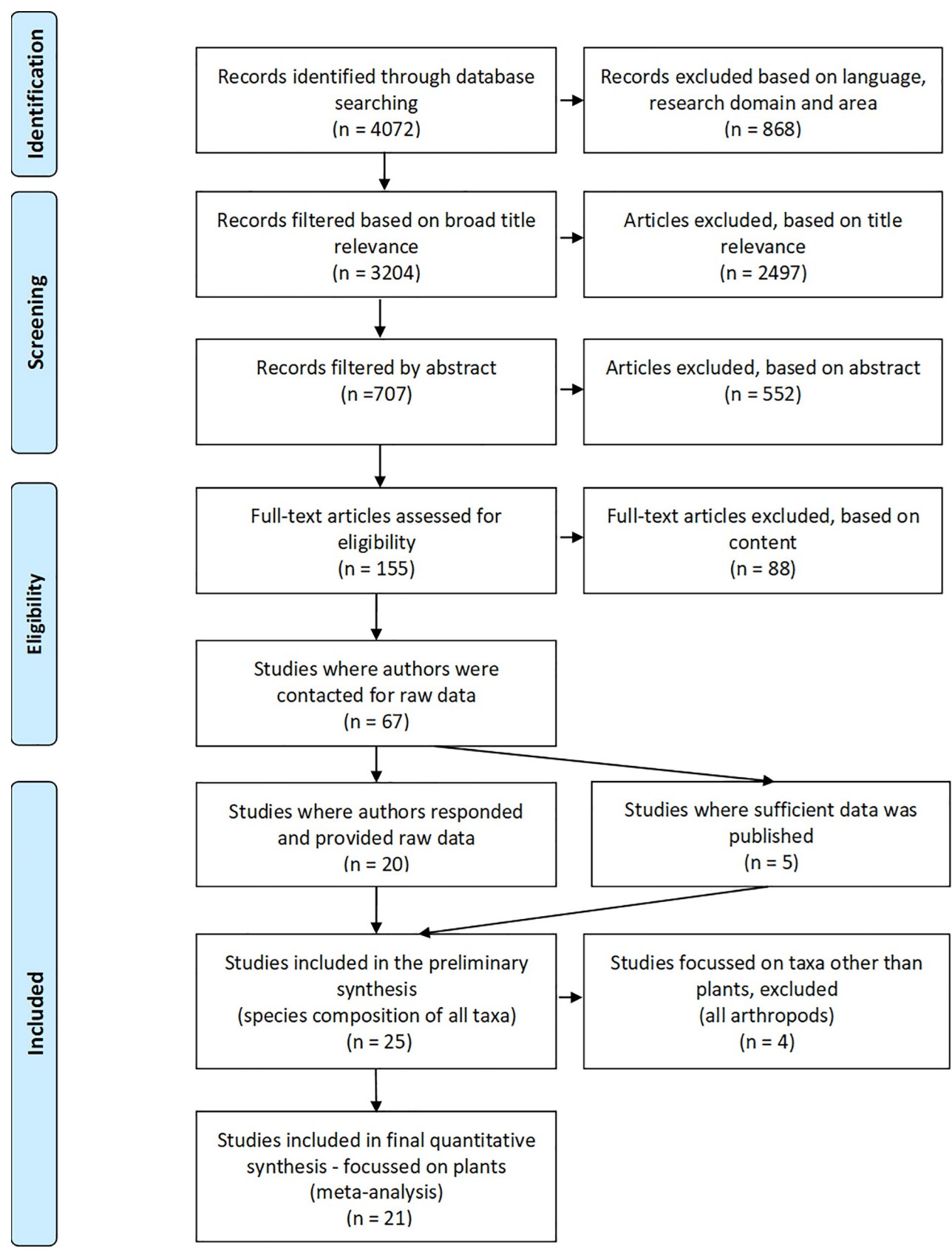

**Fig 1. A PRISMA flow diagram, depicting the process of searching for studies, filtering of studies and the inclusion of data used in this meta-analysis.** Preferred Reporting Items for Systematic Reviews and Meta-Analyses (PRISMA) flow diagram, modified from Moher *et al.* 2009 [35].

arthropods. After a preliminary analysis the four studies were excluded from the analysis in order that it focus on the single trophic level of plant species composition. Measures of ecosystem function were used exactly as reported by authors in their studies.

Studies took place in multiple countries, but were unevenly distributed among continents, with Australia, North America and Europe being well represented (ten, seven and five studies respectively) while only two studies were from Asia, one from South America and none from Africa (Table 1). The studies were also unevenly distributed among the earth's climatic biomes,

**Table 1. Data characteristics of all contributing studies used in the meta-analysis.** (Plant species composition was recorded in all sites.)

| Study | Country | Number of Restoration Sites* | Number of Reference Sites* | Degradation Type | Ecosystem Type | Ecosystem Function Types** |
|---|---|---|---|---|---|---|
| Andersen *et al.* [75] | Canada | 8 | 3 | Peat mining | Wetland | Soil litter(1), Biotic structure(1) |
| Brown *et al.* [76] | USA | 5 | 2 | Contaminated soils | Wetland | Biotic interactions(7), Soil nutrients(2) |
| Calviño-Cancela, Rubido-Bará & van Etten [77] | Spain | 25 | 5 | Clearing & plantations | Forest | Biomass(1) |
| Emery & Rudgers [78] | USA | 18 | 18 | Dune removal | Grassland | Biomass(2), Soil nutrients(4), Biotic interactions(2) |
| Forup & Memmott [79] | UK | 2 | 2 | Afforestation, agriculture | Shrubland | Biotic interactions(2) |
| Forup *et al.* [80] | UK | 4 | 4 | Afforestation, agriculture | Shrubland | Biotic interactions(2) |
| García-Palacios *et al.* [81] | Spain | (12,11) | (5,5) | Road Development | Grassland | Soil structure(1), Soil nutrients(3), Biotic Interactions(1) |
| Good *et al.* [82] | Australia | 14 | 5 | Clearing | Woodland | Biomass(3), Soil litter(1), Soil structure(1), Biotic structure(5), Soil nutrients(8) |
| Gould [83] | Australia | 31 | 36 | Mining | Woodland | Soil structure(1), Biotic structure(3), Soil nutrients(1) |
| Herath *et al.* [84] | Australia | 4 | 3 | Mining | Shrubland | Soil nutrients(8) |
| Jiao *et al.* [85] | China | (2,2) | (11,7) | Clearing | Shrubland | Soil structure(3), Biotic structure(1), Soil nutrients(6) |
| Luo, Sun & Xu [86] | China | 3 | 3 | Clearing | Wetland | Biomass(1) |
| Martin, Moloney & Wilsey [87] | USA | (1,1,1) | (3,3,3) | Agriculture | Grassland | Soil litter(2) |
| McLachlan & Bazely [88] | Canada | 28 | 3 | Clearing | Forest | Soil structure(1), Biotic structure(1) |
| Meers *et al.* [89] | Australia | 3 | 4 | Clearing & plantations | Woodland | Biotic interactions(3) |
| Miller *et al.* [90] | Australia | 2 | 2 | Mining | Shrubland | Soil nutrients(3) |
| Parrotta & Knowles [91] | Brazil | 9 | 8 | Mining | Forest | Soil litter(1), Biotic structure(1), Soil nutrients(1) |
| Polley, Wilsey & Derner [92] | USA | (5,5) | (5,5) | Agriculture | Grassland | Biomass(1) |
| Soini *et al.* [93] | Finland | 1 | 10 | Peat mining | Shrubland | Biotic structure(1) |
| Sonter *et al.* [94] | Australia | (1,1,1) | (5,5,5) | Clearing | Forest | Soil structure(1), Soil litter(1), Biotic structure(3) |
| Stefanik & Mitsch [95] | USA | 5 | 3 | Development | Wetland | Biomass(1) |

* Numbers in parentheses refer to the number of plots within distinct groups of restoration or references sites within a study (e.g. vegetation types, geographically separated areas, years etc.; treated as separate ecosystems).

** Numbers in parentheses refer to the number of ecosystem function measures in each ecosystem function type within a study.

**Table 2. Descriptions of ecosystem function types used to group the various ecosystem function measures, from all contributing studies used in the meta-analyses.**

| Ecosystem Function Type | Explanation |
|---|---|
| Biomass | Measures of live plant biomass or primary productivity. |
| Biotic Structure | Structural characteristics of the plant community such as total plant cover, tree height and canopy cover/volume. |
| Biotic Interactions | Complex interaction between species, or between species and the environment. These interactions may relate to propagation of groups of species within the ecosystem (e.g. pollination, animal facilitated seed dispersal and seedbank composition/viability) or presence of important biota (e.g. soil invertebrates, bacteria and fungi) which fulfil multiple roles in the ecosystem (e.g. decomposition, soil aeration, mutualisms with plants). |
| Soil Litter | Measures of leaf litter and other dead plant material (e.g. dead wood) on the soil surface, but excludes measures of decomposition. |
| Soil Nutrients | Measures of nutrient pools in the soil, as well as indicators of nutrient cycling. |
| Soil Structure | Measures related to soil temperature, stability, texture, and water retention. |

with tropical biomes being the most poorly represented, but were fairly evenly distributed among five ecosystem types (based on vegetation physiognomy): forest, woodland, shrubland, grassland and wetland. Measurements of ecosystem function were fairly evenly distributed into six broad categories, three measuring biological functions: *biomass*, *biotic structure*, *biotic interactions*, and three measuring soil functions: *soil litter*, *soil nutrients*, *soil structure* (Table 2).

For the analysis of the level of variation in intact ecosystems there were 28 different groups of reference sites. This produced 28 measures of species composition similarity (because some studies had reference sites in multiple ecosystems) and 55 measures of ecosystem function similarity (because some studies measured multiple ecosystem function types). For the analyses comparing restoration sites to intact ecosystems, the 21 studies yielded a total of 1850 unique measures, which allowed 478 comparisons of species composition and ecosystem function means (mean response ratios) among 205 restoration sites (again because most studies reported on multiple restoration sites and measured multiple ecosystem function types). (S1 Dataset) contains the full dataset, and (S1 Checklist) the PRISMA checklist [35].

## Similarity metrics and response ratios

Similarity metrics were used to compare similarity in species composition among sites. The Bray Curtis metric has a number of numerical qualities which makes it is especially suitable for comparing species composition among ecological communities. For example, it can accommodate different measures of species profusion (e.g. counts or abundance, cover biomass and density) and ignores "joint absences" so does not consider samples similar just because they both lack a certain species [36]. Data for the abundance, cover, biomass etc. of each species were first squared root transformed to down weight the importance of overabundant species [36]. Similarity matrices were constructed and analysed using PRIMER v. 6 [37].

Given that ecosystems always display some level of heterogeneity, variation will inevitably exist among samples from two or more reference sites within an ecosystem. By using only restoration studies which had multiple reference sites we were first able to compare the average variability among reference sites within a study ecosystem in terms of both species composition and ecosystem function. This metric (mean % similarity between reference sites) was then used as a baseline with which to gauge the similarity of restoration sites to the range of recorded states for the 'intact ecosystem'. Thus creating a metric that allows us to examine the

relationship between species composition and ecosystem function within restoration sites that is comparable across all the different ecosystems reported in the restoration studies.

To derive the mean % similarity for species composition among the reference sites for each study, we used measures of species profusion (e.g. abundance) to construct pairwise Bray-Curtis similarity matrices. First, using all pairwise values for reference sites, we calculated a mean % similarity for reference sites within each ecosystem. If a study included logical groupings (e.g. spatially segregated groups of reference and restoration sites) we treated these as separate ecosystems. Next, we compared the similarity of each restoration site to the range of the relevant reference sites. To do this we used the same pairwise Bray-Curtis metrics, with the comparison this time being between a restoration site and each of its reference sites. These comparisons were then used to derive a mean % similarity for each restoration site relative to the reference sites in an ecosystem.

Using both the mean % similarity within references sites and the mean % similarity between the restoration and reference sites, we were then able to calculate a response ratio which explicitly evaluates how close a restoration site is to the range of reference sites in a study. To do this we modified the traditional response ratio [38], $ln(RR+1) = ln(REST+1/REF+1)$, where REST is the mean % similarity of restoration sites to reference sites and REF is the mean % similarity within reference sites. To account for zero values we added a value of one to both the numerator and denominator [e.g. 39].

The same response ratio was also calculated for ecosystem function. However, unlike species composition (where the Bray-Curtis metric was used to reduce multi-dimensional data to a single comparative metric), each site had only one value relating to an ecosystem function. Therefore, to calculate the pairwise similarities of ecosystem function, for both among reference sites and between restoration and reference sites, we simply used the ratio of the smaller to the larger measure. In cases where multiple measures of ecosystem function within the same ecosystem function type were provided, the mean response ratio across all those functions was used (rather than each measure separately) to avoid the analysis being unduly weighted by numerous measures from a few studies or from a few ecosystem function types.

## Data analysis

First, we tested whether there was more similarity, within intact ecosystem sites, in species composition or in ecosystem function, using a general linear model. The response variable was the mean similarity within reference sites (as described above), and the explanatory variable was the type of measure (ecosystem function or species composition). Secondly, for ecosystem function we tested whether this similarity differed between different ecosystem function types, and for species composition we tested whether the similarity differed between ecosystem types. In this second model, the response variable was either the mean similarity within reference sites in terms of ecosystem function or species composition, and the explanatory variable was either the function type or the ecosystem type.

General linear mixed models were then used, with data from restoration sites, to assess the relationship between species composition and ecosystem function. The base model used the ecosystem function response ratio as the response variable and species composition response ratio as a fixed explanatory covariate. Two random terms, 'study' and 'ecosystem function type', were also included to account for multiple and differing numbers of restoration sites in each study and also to account for non-independence of multiple ecosystem function measures at some sites.

We explored whether ecosystem type or ecosystem function type influenced the relationship by including these factors and their interaction with species composition as fixed effects in the model, and 'study' retained as a random term. Backwards stepwise selection was used,

starting with all predictor variables included in the model and removing factors and evaluating their influence on corrected Akaike Information Criterion (AICc). The Best-fit model chosen was the one with the lowest AICc value. If models provided comparable AICc values (within 2 units of the best model), then the one containing the fewest variables was chosen (S2 Dataset) contains the full list of AICc values. The models were fitted with the maximum likelihood (ML) criterion to allow comparison using AICc, but to obtain parameter estimates the models were refitted with restricted maximum likelihood (REML) criterion. The significance of the predictor variables in final models were examined using Type III F-tests. The Kenward-Roger approximation was used to estimate the denominator degrees of freedom and calculate p-values. Preliminary analyses tested additional factors in general linear mixed models (years since restoration commenced and active vs passive restoration). These factors did not increase resolution in the model, but reduced their power and scope as not all studies reported data for these factors (S3 Dataset) describes the models used in the final meta-analysis.

The nature of any interaction was explored by re-running models for a subset of data for each different group (i.e. each ecosystem function type). In addition to generating scatterplots of response ratios, values and model outputs were back-transformed to positive numbers 0–100% to be more intuitive to understand. Positive response ratios (values over 100%) were assigned the value of 100% (as they were within the range of intact ecosystems).

For all models, model fit was assessed visually using the residual and q-q norm plots to ensure model assumptions were not violated. To examine the proportion of the variance explained by the models we used the approach of Nakagawa & Schilzeth's [40] to generate the marginal $R^2$ (fixed effects alone—considered analogous to the $R^2$ value used in simple linear models) and the conditional $R^2$ (fixed and random effects). All analyses were conducted in R [41].

## Results

### Heterogeneity within intact ecosystems

Among reference sites there was considerable range in the similarity of both plant species composition and ecosystem function (species composition ranged from 23 to 88%, ecosystem function ranged from 27 to 98%). Similarity was, however, greater for ecosystem functions than for species composition ($F_{1,81}$ = 26, p >0.001, Fig 2). Thus, these results indicate that intact ecosystems were more variable in terms of species composition than ecosystem function. The degree of similarity in ecosystem function did not differ between different ecosystem function types ($F_{5,49}$ = 0.76, p = 0.59, Fig 3) nor did the degree of similarity in species composition differ across different ecosystem types ($F_{4,23}$ = 1.7, p = 0.19, Fig 3).

### The species composition—ecosystem function relationship

Overall we found a positive relationship between plant species composition and ecosystem function indicating that as a site's species composition is restored (i.e. as it becomes more similar to references sites) so is its ecosystem function (Fig 4).

The base model's positive relationship between species composition and ecosystem function was significant, but only explained a small amount of the variance in the dataset. The marginal $R^2$ ($R_M^2$) was only 2%, indicating that the fixed effect alone (species composition) explains very little of the variance in ecosystem function (Table 3). When back-transformed and plotted on a 0–100% scale, the relationship was curvilinear, and although weak, indicated a positive saturating curve, that even at full species composition does not attain the ecosystem function levels of intact ecosystems (Fig 5).

Our best fitting model included species composition, ecosystem function type and their interaction, but excluded ecosystem type. This indicates that the inclusion of ecosystem type had little

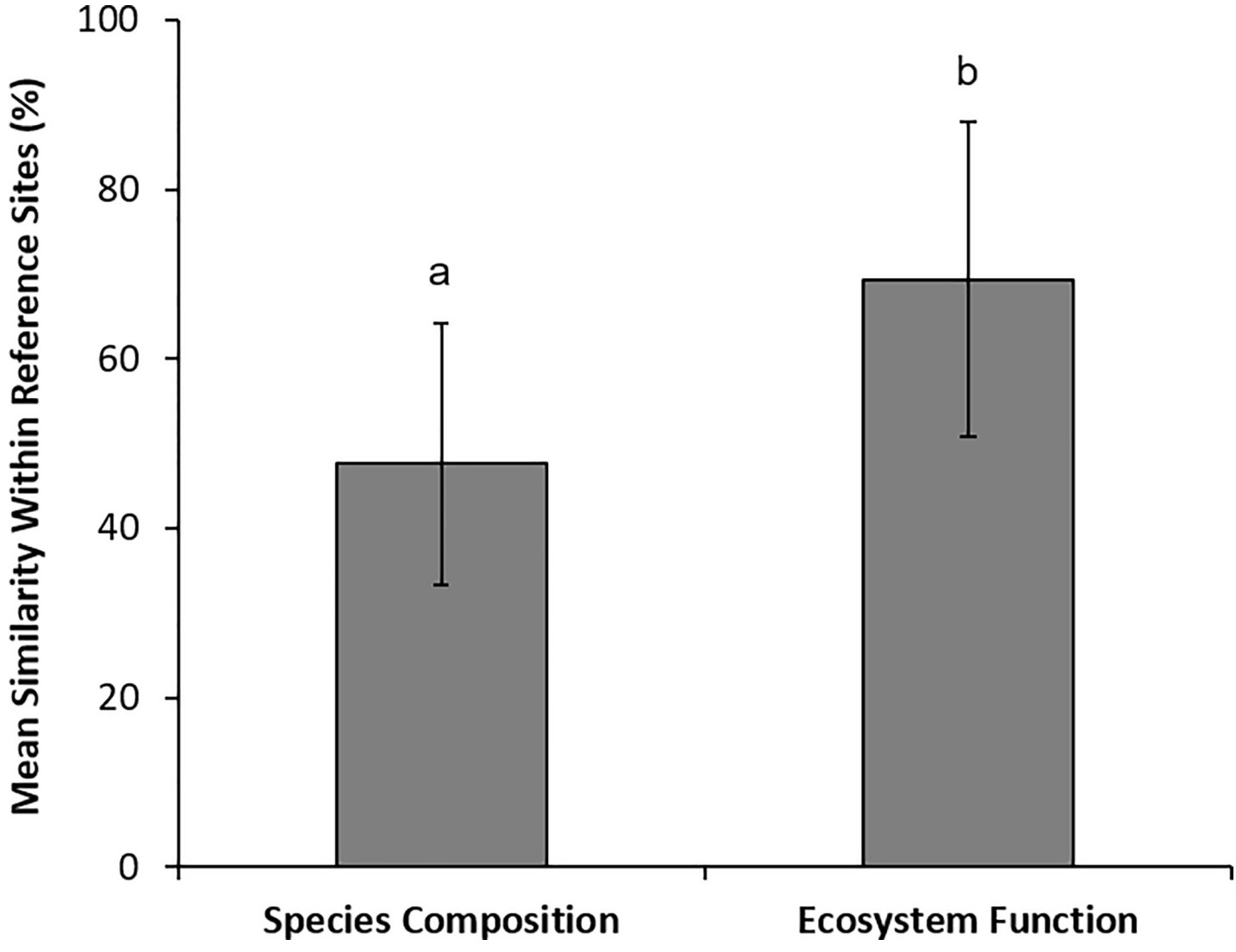

**Fig 2. Mean similarity (±SD) in species composition and ecosystem function within intact natural ecosystems globally (i.e. reference sites: n = 28 for species composition, n = 55 for ecosystem function).** Similarity for species composition is the mean of pairwise Bray-Curtis similarity measures for each group of reference sites within a study, and similarity for ecosystem function is the mean of pairwise ratios for each ecosystem function type in each group of reference sites within a study. A significant difference (p < 0.05) between the mean similarity of species composition and the mean similarity of ecosystem function was found using a general linear model, and is denoted by different letters.

impact on the relationship, and that the relationship between species composition and ecosystem function was not consistent among different ecosystem function types. The best-fit model had far greater explanatory power than that of the base model ($R_M^2$ = 21%; Table 3). A preliminary analysis using the same model, but including data for from four arthropod studies together with the 21 plant species composition studies, had almost identical results ($R_M^2$ = 21%; $R_C^2$ = 33%).

The interaction between ecosystem function type and species composition was explored by examining the model outputs for each ecosystem function separately (Fig 6). Species composition was only significantly associated with two ecosystem function types (Table 4, Fig 6). For the functions *biomass* and *biotic structure*, species composition explained a sizable amount of variance in the data ($R_M^2$ = 31% and $R_M^2$ = 17% respectively, Table 4), and when back-transformed to a 0–100% scale these two ecosystem functions exhibited strong saturating

**(A)**

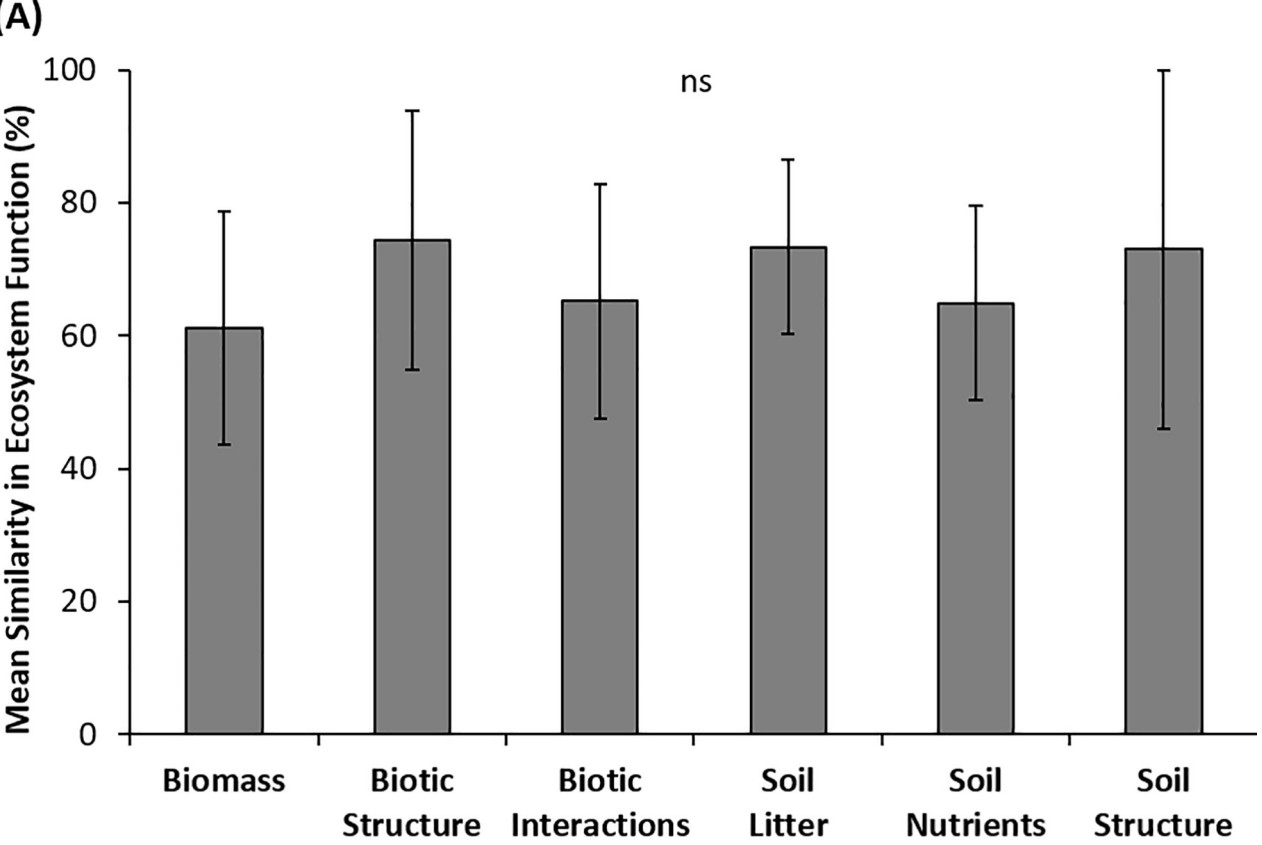

**(B)**

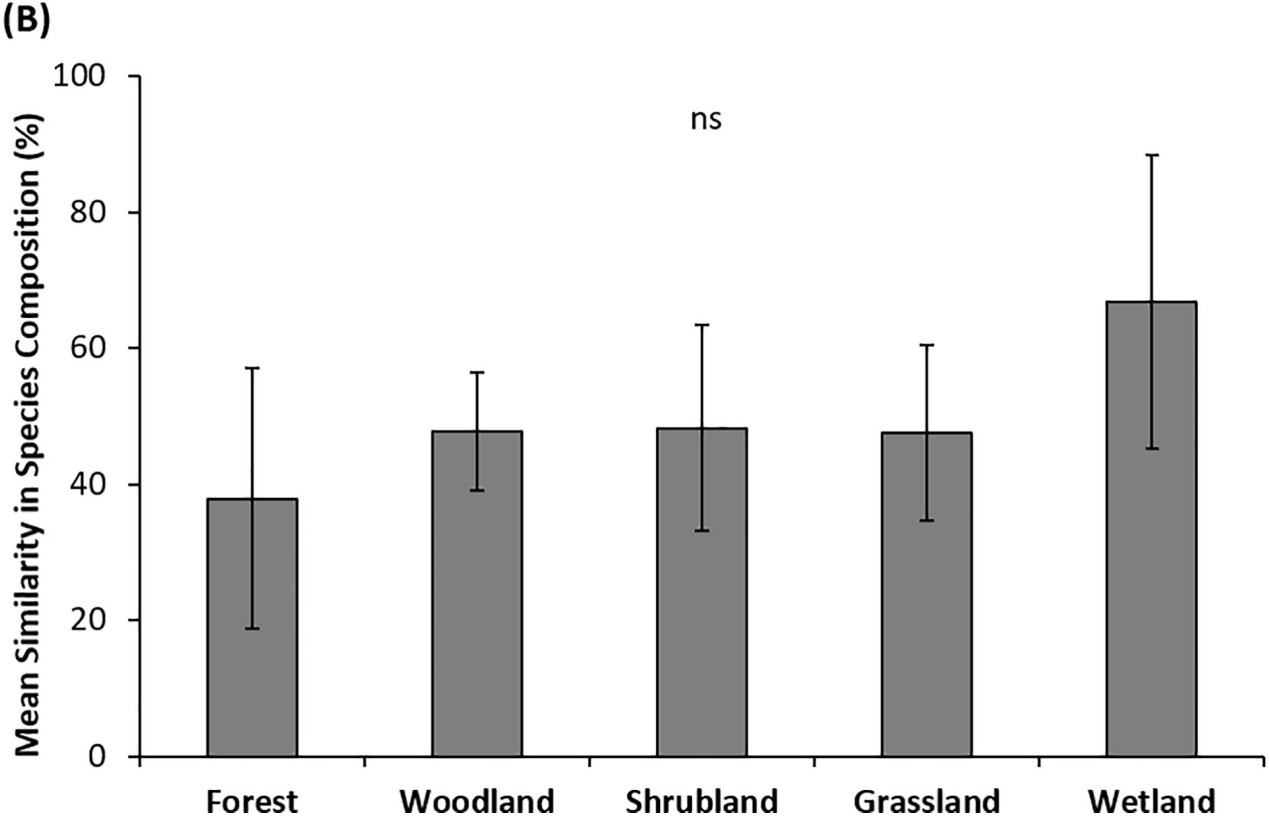

**Fig 3.** (A) Mean similarity (±SD) in ecosystem function among different ecosystem function types within intact natural ecosystems globally (i.e. reference sites). (B) Mean similarity (±SD) in species composition (Bray-Curtis similarity) among different ecosystem types globally (i.e. reference sites). Similarity is the mean of pairwise ratios within each group of reference sites within a study, for each function or ecosystem type. There were no significant differences among function types or among ecosystem types.

relationships (Fig 7). For the other four ecosystem functions types (*biotic interactions*, *soil litter*, *soil nutrients* and *soil structure*), the relationship between species composition and function were not significant, most of the points sitting at or close to the level of functions in intact ecosystems regardless of species composition, indicating that the identity of species in the ecosystem was immaterial to those ecosystem functions.

## Discussion

### The species composition—ecosystem function relationship

In this study our interest was not simply in whether adding or removing species to an ecosystem allows us to detect changes in one or other ecosystem function [e.g. 42]. Instead, we are interested in concomitant changes in the levels of both species composition and ecosystem

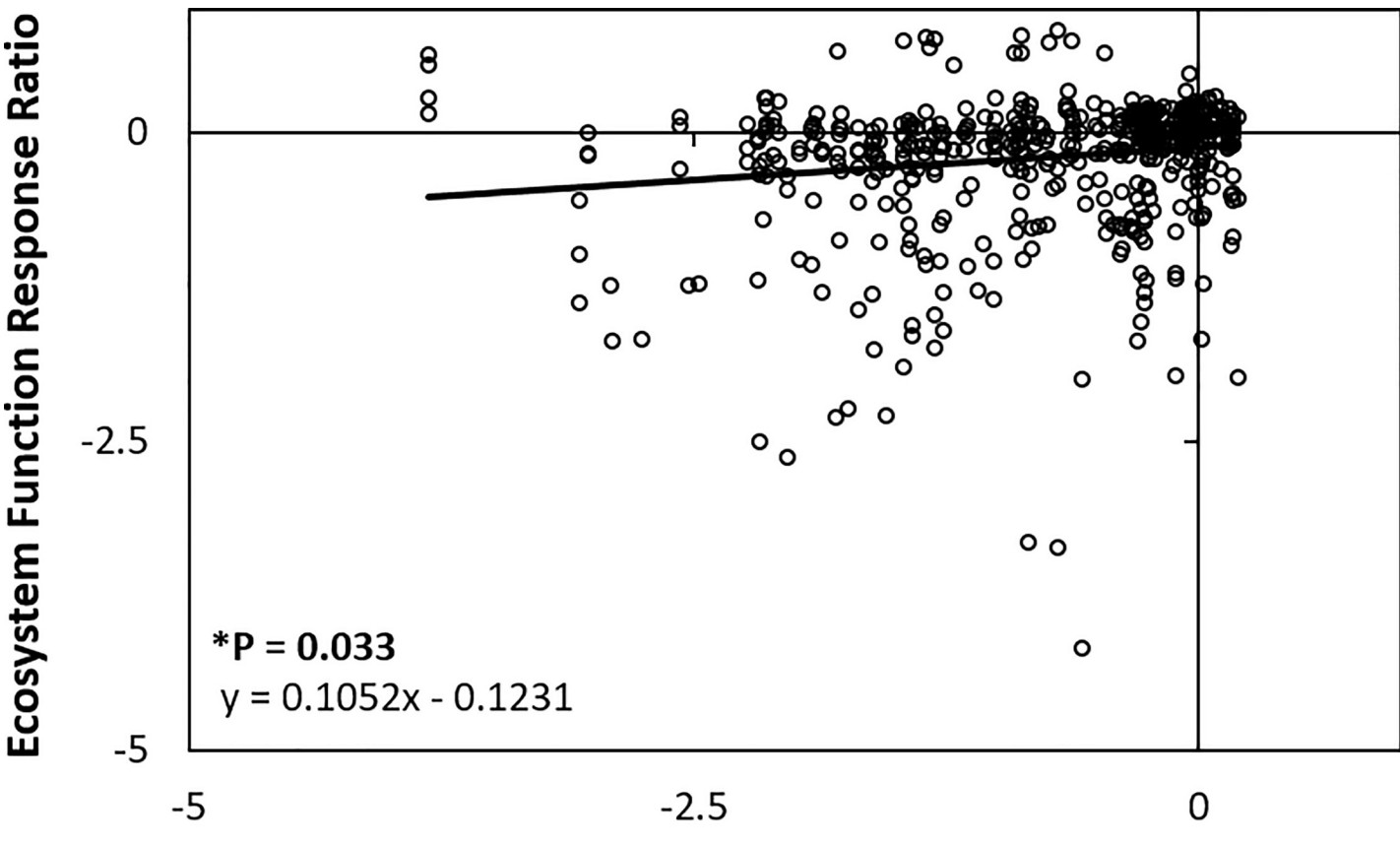

**Fig 4. Relationship between species composition and ecosystem function in 478 mean measures from 205 restoration sites, in 21 studies around the globe.** For each measure, for species composition or ecosystem function, the response ratio is a measure of how similar the restoration sites are to their reference sites, and has the formula ln(similarity of restoration sites to reference sites +1/similarity within reference sites + 1). Values ≥ 0 on the x-axis or the y-axis can be considered to be restored to within the range of intact ecosystems (i.e. reference sites). The line is the output from a general linear mixed model with ecosystem function type and study as random factors. * Denotes a significant relationship.

**Table 3. Output of the general linear mixed models for the relationship between species composition and ecosystem function, in 21 studies around the globe.** Analysis includes the base model which controls for the random effects of study and ecosystem function type and the best-fitting model which includes ecosystem function type and the interaction between species composition and ecosystem function type as fixed factors, and study as a random factor. SC = Species Composition; EFtype = Ecosystem Function Type (for descriptions see Table 2).

| Model and Component | F-value | *df | *p-value | Deviance explained (%) | **$R_M^2$ (%) | **$R_C^2$ (%) |
|---|---|---|---|---|---|---|
| **Base Model:** | | | | | | |
| SC + (study + EFtype) | | | | | 1.98 | 35.5 |
| SC | 4.6 | 1, 245 | 0.033 | 0.7 | | |
| **Best-fitting Model:** | | | | | | |
| SC +EFtype+ SC*EFtype + (study) | | | | | 21.0 | 35.1 |
| SC | 14.3 | 1, 266 | <0.001 | 0.7 | | |
| EFtype | 2.9 | 5, 402 | 0.01 | 11.7 | | |
| SC*EFtype | 4.7 | 5, 398 | <0.001 | 3.3 | | |

* The Kenward-Roger approximation was used to estimate the denominator degrees of freedom (numerator, approximated denominator) and calculate p-values.

** Estimated variance is explained by marginal $R^2$ values ($R_M^2$ = fixed factors only) and conditional $R^2$ values ($R_C^2$ = both fixed and random factors).

function relative to their intact natural condition. BEF research has often failed to incorporate natural levels of diversity and heterogeneity, and consequently studies have frequently been conducted in artificially simplified ecosystems [7,24]. Even when these studies use natural ecosystems, they tend to be conducted in ecosystems with inherently low levels of diversity or those with relatively simple structure, particularly grasslands [12,42,43]. The concentration of studies on low diversity ecosystems makes extrapolating to more complex ecosystems problematic [6,44,45]. Our approach overcame many of these inherent problems and allowed us to test the generalisable nature of these relationships across different ecosystems, regardless of their inherent level of biodiversity (i.e. species rich or species poor ecosystems). In doing so we ensured that these relationships were directly relevant to real-world ecosystems.

Our global meta-analysis of ecosystem function in relation to species composition across a range of degraded, recovering and intact ecosystems revealed the following generalities (contrast with the generalities in BEF studies outlined in the Introduction):

1. The overall relationship between species composition and ecosystem function is positive. This demonstrates that the types, and abundance, of species present in an ecosystem can influence how an ecosystem functions.

2. The relationship between species composition and ecosystem function is non-linear and saturating, but it was not consistent across all the ecosystem functions that were explored.

3. Different ecosystem functions exhibited different relationships with species composition, and for some functions we found no relationship at all. Consequently, the weak relationship in the base model was strengthened by an order of magnitude when the type of ecosystem function is taken into account.

4. We did not explore the relationship between species composition and ecosystem multifunctionality. Although analytically complex, this may be a rewarding avenue for future studies utilising the rapidly expanding data available from restoration ecology and similar fields.

5. We did not test the stability of ecosystem function relative to species composition, and consider it unfeasible using this type of data (as it entails holding species composition constant but lower than intact ecosystems, across time or space, in order to generate the replicate measures needed to generate reliable stability metrics across a range of species compositions).

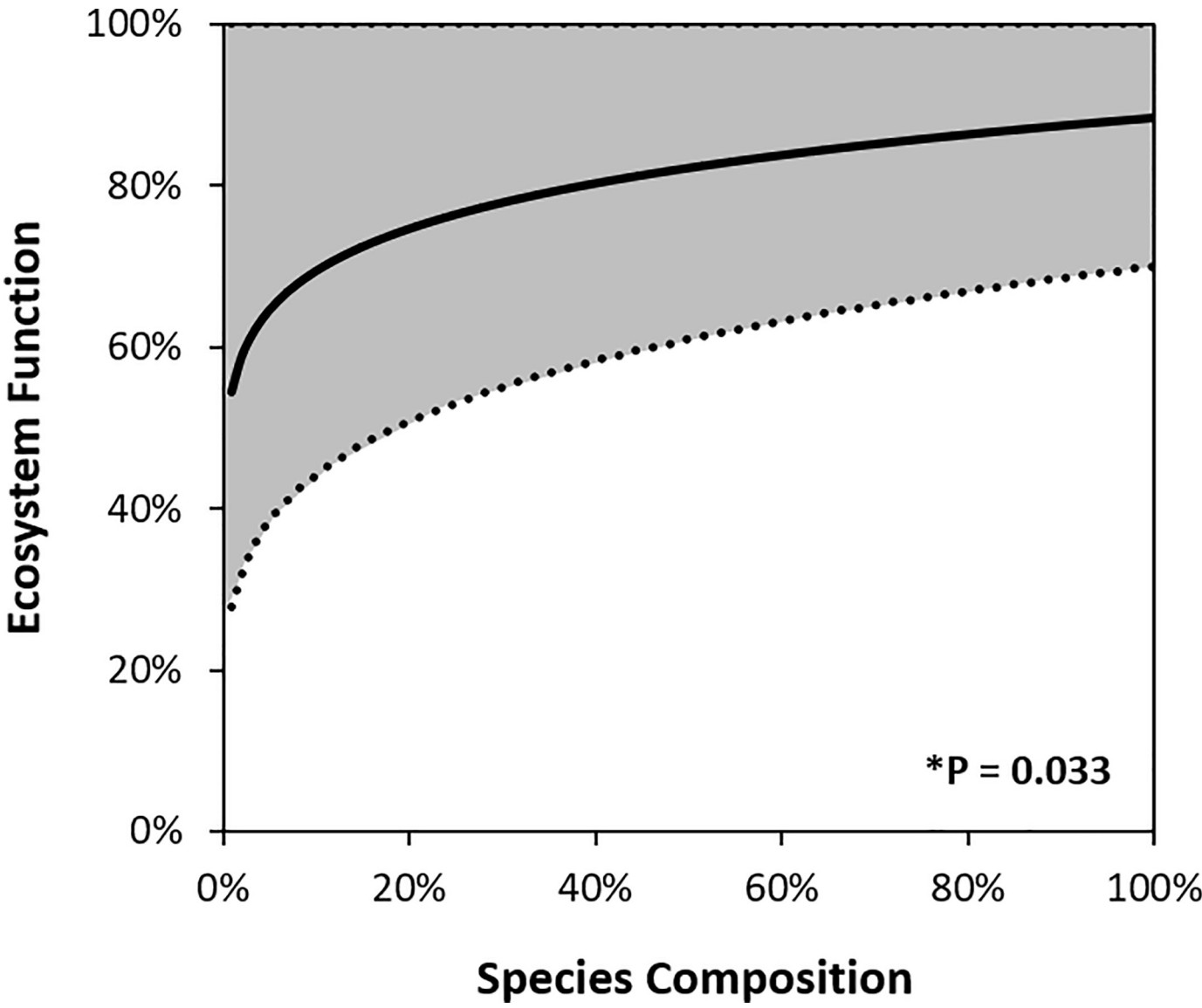

**Fig 5. Modelled relationship between species composition and ecosystem function, in ecosystems recovering towards intact conditions.** Data have been back-transformed into positive values by taking the exponent of both the species composition and ecosystem function response ratios. The black line is the model output and the grey area represents 95% confidence intervals around the model. * Denotes a significant relationship.

Unlike the differences in the type of ecosystem function, accounting for differences in the type of ecosystem did not affect the relationship between species composition and ecosystem function in our models, implying that the relationship may well be generalizable across global ecosystems. The number of studies across the different ecosystem types in our meta-analysis was small and thus our ability to identify differences in the relationship among ecosystem types was fairly weak. Aquatic and terrestrial ecosystems have been found to have similar species richness–ecosystem function relationships in a number of studies [4–6,15,17]. Our meta-analysis, similarly, did not differentiate wetland from the other four terrestrial ecosystem types. Few BEF meta-analyses compare species richness–ecosystem function relationships among terrestrial ecosystem types, but in the comprehensive study of Cardinale *et al.* [6] results were also fairly consistent across ecosystem types, with the only difference being a

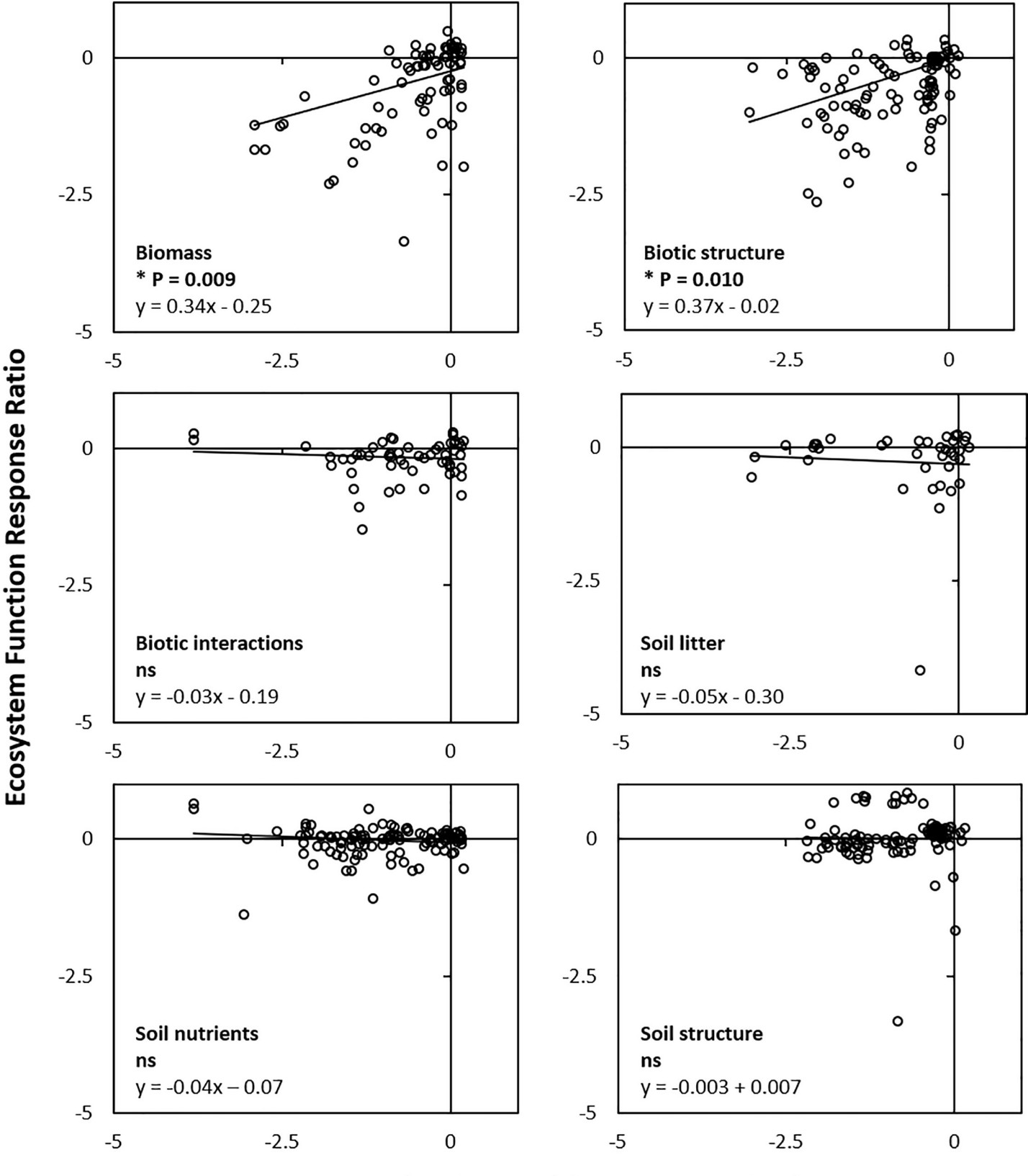

**Fig 6. Relationship between species composition and ecosystem function for each of the six ecosystem function types, for 478 mean measures from 205 restoration sites, in 21 studies around the globe.** For species composition or ecosystem function, the response ratio is a measure of how similar the restoration sites are to their reference sites, and has the formula ln(similarity of restoration sites to reference sites +1/similarity within reference sites + 1). Values ≥ 0 on the x-axis or the y-axis can be considered to be restored to within the range of intact ecosystems (i.e. reference sites), and lines are the output of separate general linear mixed models. * Denotes a significant relationship.

suggestion that forest ecosystems responded differently in terms of biomass functions. In the smaller Balvanera etal. study [4] the relationships were actually weaker in ecosystem types with more studies (grassland, forest, aquatic and marine), and stronger in ecosystem types for which there was fewer data (ruderal, crop, salt marsh, bacterial and soil).

In common with BEF studies, our results are not drawn evenly from continents and biomes around the world. Temperate grasslands dominate species richness–ecosystem function studies [4,6,8,46], and large meta-analyses have repeatedly found that tropical biomes and the continents of South America and Africa are underrepresented [4,6,8,46,47]. Our study reflects a similar bias in restoration research across the globe.

## Nature of species composition and other biodiversity relationships

BEF research has been one of the most prominent areas of ecological research over the past three decades [7]. Even in highly simplified and microcosm experiments, however, results have not always been consistent [4–6,8] and the explanatory power of fitted relationships between species richness and ecosystem functions has ranged widely (e.g. $R^2$ = 71% [5]; $R^2$ = 29–73% [6]). Attempts to generalise across real ecosystems have produced significant relationships with ecosystem function, but also found large variance [46,48,49]. For example, Maestre *et al.* [49] explored the relationship between species richness and productivity/nutrient functions in multiple drylands across the globe. They found that species richness was ranked amongst the best predictor variables for ecosystem function, although on its own it accounted for very little of the variation (highest $R^2$ value = 3.2%). In two recent global meta-analyses, using data from naturally assembled communities, abiotic factors and functional composition were found to be stronger drivers of ecosystem function than species richness [46,50]. In our study, the weak relationship between species composition and ecosystem function suggests factors other than species composition may control the recovery of ecosystem function.

Being able to reliably predict the point of biodiversity change where large or irreversible damage to ecosystem function occurs also remains elusive [6,7]. Expert opinion originally estimated that 50% of species would be required to maintain ecosystem functions at 75% of their maximum [51]. More recent analyses have suggested this may be an underestimate [6].

**Table 4. Output of general linear models for each of the six ecosystem function types separately.** Each model contains species composition as a fixed factor and study as a random factor. (For descriptions of ecosystem function types see Table 2.)

| Ecosystem Function Type | F-value | *df | *p-value | Deviance Explained (%) | **$R_M^2$ (%) | **$R_C^2$ (%) |
|---|---|---|---|---|---|---|
| Biomass | 7.260 | 1, 65 | 0.009 | 6.5 | 31.4 | 33.2 |
| Biotic Structure | 7.250 | 1, 40 | 0.010 | 5.7 | 17.2 | 49.4 |
| Biotic Interaction | 0.190 | 1, 23 | 0.666 | 0.4 | 0.7 | 20.0 |
| Soil Litter | 0.060 | 1, 3 | 0.823 | 0.3 | 0.4 | 6.2 |
| Soil Nutrients | 0.790 | 1, 63 | 0.377 | 0.7 | 1.4 | 38.4 |
| Soil Structure | <0.001 | 1, 11 | 0.984 | <0.1 | <0.1 | 8.8 |

* The Kenward-Roger approximation was used to estimate the denominator degrees of freedom (numerator, approximated denominator) and calculate p-values.

** Estimated variance is explained by marginal $R^2$ values ($R_M^2$ = fixed factors only) and conditional $R^2$ values ($R_C^2$ = both fixed and random factors).

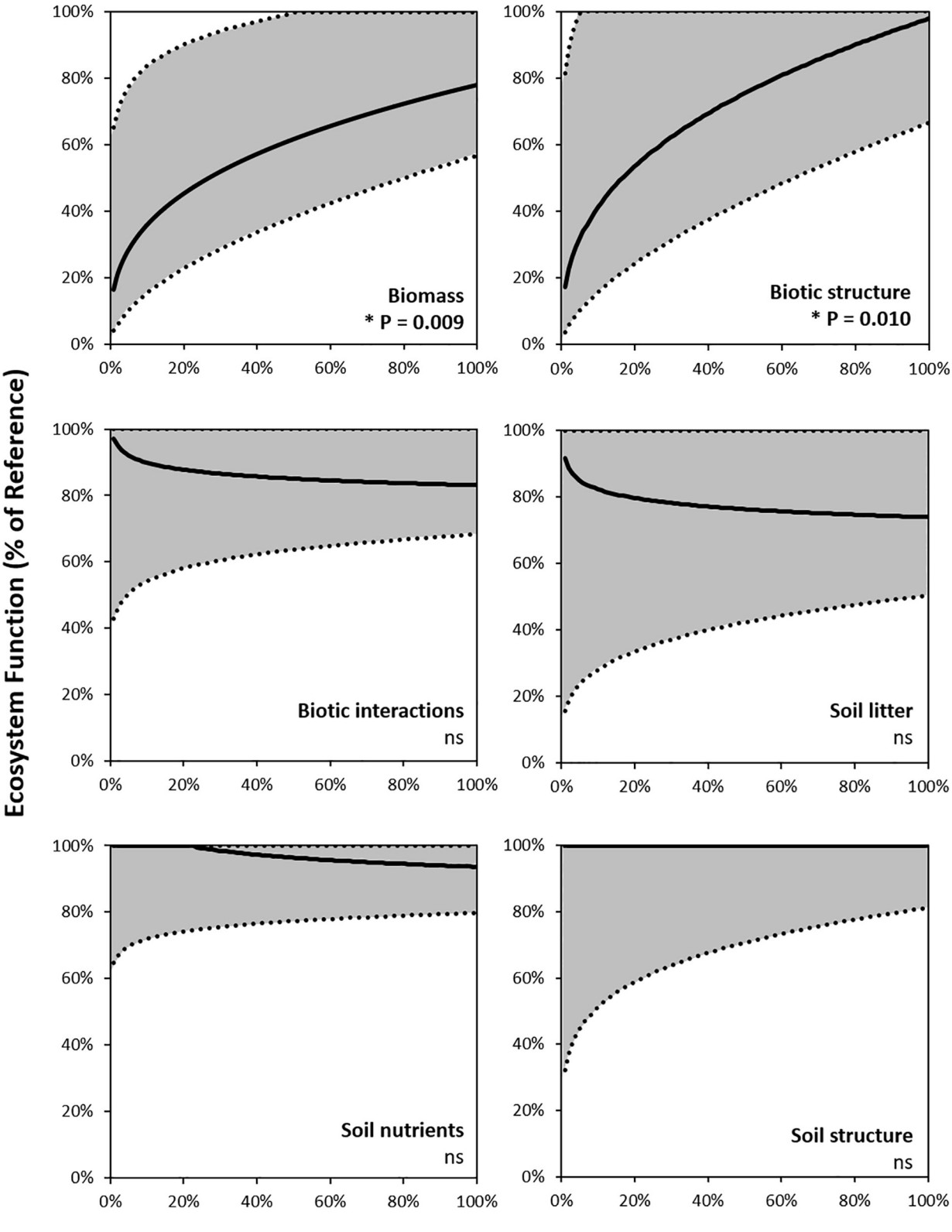

**Fig 7. Modelled relationship between species composition and ecosystem function for each of the six ecosystem functions types, in ecosystems recovering towards intact conditions.** Data have been back-transformed into positive values by taking the exponent of the response ratio. The black line is the model output and the grey area represents 95% confidence intervals around the model. * Denotes a significant relationship.

Perhaps a more pertinent question, and one that we have attempted to address here, is what proportion of the species composition is required to maintain ecosystem function at similar levels to those of intact ecosystems? Our results suggest that on average, species composition would need to be 40–50% similar in order to have functions at 75% of reference site means. The major increases in ecosystem function, as a whole, occurred within the first c. 20% of similarity in species composition. These estimates provide a starting point for further exploration of species composition relationships. However, the high levels of variation and inconsistency of the relationships among the different ecosystem function types do limit the generality of these predictions.

These studies suggests that a few species may be responsible for the majority of functioning within an ecosystem, with additional species providing limited further benefits. If this is consistently true, the implications for biodiversity are substantial, but we cannot on this basis, claim that dramatically reduced species composition would be adequate in providing ecosystem functions in most ecosystems, under most conditions or most of the time. In fact this suggested redundancy may be the primary mechanism that confers stability and resilience to ecosystems, ensuring that ecological functioning is maintained despite the decline or extinction of particular species or the change in conditions within an ecosystem, and this has been termed the insurance effect [2,14]. This insurance effect is especially relevant when larger space and time scales are considered [1,52,53]. Isbell *et al.* [53] have comprehensively debunked a simplistic view of redundancy, showing clearly that the proportion of species in an ecosystem providing an ecosystem function, increases with the number of years, places and environmental changes considered, and that these increases are further compounded by interactions between these factors, supportive of a general complementarity rather than simple redundancy.

There is overwhelming evidence that species richness and species composition play a role in determining ecosystem function [4–8], but if this role only accounts for a small proportion of the variance in most real-world ecosystems, then the emphasis given to this relationship should be re-evaluated. There is a need to examine other factors that play a role in ecosystem function, if we intend to ensure their sustainability [6].

## Heterogeneity within intact ecosystems

The consistently low levels of similarity in species composition that was found within intact ecosystems, highlights that there are many naturally occurring combinations of species occupying any one ecosystem. Despite the resurgent attention on alternate stable states in ecology [54,55] the levels of species heterogeneity inherent in natural ecosystems has typically been underestimated. This is implicitly demonstrated by the fact that many studies that report on compositional change in ecosystems, ascribe the change to an external impact (e.g. changes in climate, fire, herbivory etc.), rather than imagining compositional drift to be an inherent dynamic.

The levels of similarity in species composition among intact ecosystems reported here were low (a mean of 48% in Bray-Curtis similarity) but were similar to other studies with comparable statistics, measured through time rather than space (e.g. fynbos heathlands in South Africa [56,57], jarrah forests in Australia [58], upland grasslands in Wales [59] and semi-arid

succulent karoo shrublands in South Africa [60]). Across a broad range of ecosystem types therefore, without external impacts beyond natural disturbances, it is not unusual for half the species composition to change within a single site, over about 30 years.

## Differences in the relationship among ecosystem function types

Surprisingly only two of the six ecosystem function types exhibited strong and significant relationships with species composition, namely, *biomass* and *biotic structure*. Measures of biomass and productivity are ecosystem functions most commonly used in BEF research, and are invariably found to have among the strongest relationships with species richness in meta-analyses [1,4,6,7,9,10,61,62]. For biomass functions, however, even when species composition was fully recovered, ecosystem function remained lower than that of intact sites, indicating that full species composition alone was not sufficient to attain full ecosystem functioning. These functions are often heavily influenced by large, slow-growing plant species, which need to reach a certain size before full levels of these functions are achieved [63]. Even when not governed by specific large species, some functions may only fully return with the passing of time [e.g. 45,64–66].

More that half of the *biotic interactions* concerned soil-based interactions (e.g. bacteria, fungi and biological crusts) and the flat species composition relationship of this function is consistent with those of the other soil-based ecosystem functions in our meta-analysis. While measures of *soil structure* are not widely reported in the BEF literature, measures of soil nutrient pools, mineralisation and decomposition are, and meta-analyses frequently find these *soil nutrient* and *soil litter* functions to have a weak but significant relationship with species richness [1,4,6,7,9,10,62]. Potentially then species richness and species composition relationships may differ for these functions The inference being that vastly different, but diverse, species compositions may all support the development of a certain level of nutrient cycling and availability.

The ecosystem function concept is broadly defined, with some ecosystem functions apparently not driven by either species richness or composition [4,6,7,45,50]. The field as a whole would benefit from a greater refinement of the concept.

## Implications for restoration ecology

The high level of variation found within intact reference ecosystems in our study emphasises the importance of including multiple reference sites against which to compare any altered ecosystem. The field of restoration ecology has recognised both that vegetation may occur naturally in a range of species compositions, or states [67,68], and the corollary that there is a need to use multiple reference sites in restoration projects [28,30,31]. Without a baseline which captures the inherent heterogeneity of the broader target ecosystem it is very difficult to accurately assess whether a site should be considered intact, degraded or on a path of recovery between the two. The end goal for restoration projects should not be a specific reference point but rather any point within a cloud of reference conditions or states. The reference conditions can be defined both in terms of species composition and in terms of ecosystem function, with our results suggesting more inherent heterogeneity expected in species composition measures than in ecosystem function measures.

Our results also suggest that restoring species composition cannot be taken as a proxy for restoring ecosystem function, or vice versa. Restoring ecosystems for function alone, can lead to the assembly of novel ecosystems which do not resemble the reference sites' species composition [69]. This may be misaligned with conservation goals, especially if novel ecosystems contain exotic or invasive species [69,70]. While we would not consider an ecosystem containing

significant invasive species restored, it is necessary to accept that, accelerated by multi-faceted human-induced global change, it may no longer be possible to reinstate specific assemblages if, for instance, component species have shifted range, have been extirpated [71] or the underlying substrate, rock structure or hydrology has been altered. Developing goals for recovering critical levels of ecosystem function should concern restoration ecologists as much as recovering critical levels of ecosystem composition.

Some ecosystem functions may be more easily restored than others. Not all ecosystem functions are equal measures of ecosystem recovery, and in some cases, are not indicative of recovery at all. Therefore understanding the role and sequence that functions play in the trajectory of restoration is critical [72,73]. For example, monitoring *soil nutrients*, *soil structure*, and potentially soil *biotic interactions*, may be critical in the early stages of restoration, as their recovery may be an obligate condition for the restoration of the ecosystem as a whole, but the emphasis may shift to biological functions at later stages of restoration [63–68]. There may be a variety of goals and priorities for each restoration project, but a project should never focus solely on any single component of the ecosystem. A true test of restoration efficacy would be to target the functions that are hardest, rather than easiest, to return [74].

## Supporting information

**S1 Checklist. PRISMA (preferred reporting items for systematic reviews and meta-analyses) checklist to assist in the critical appraisal of the meta-analysis and systematic reviews.**
(DOC)

**S1 Dataset. Data and similarity metrics for each restoration site (n = 205), including each measure of ecosystem function.**
(XLSX)

**S2 Dataset. Corrected Akaike Information Criterion (AICc) for each general linear mixed model.**
(XLSX)

**S3 Dataset. Descriptions of the models run.**
(XLSX)

## Acknowledgments

The authors would like to thank Arjun Amar for statistical advice and comments on earlier versions of this paper. The authors would also like to thank Jasper Slingsby and a number of anonymous reviewers for comments that greatly improved the paper. We are particularly grateful to all the authors of the studies used for providing their raw data, without which we could not have undertaken this study.

## Author Contributions

**Conceptualization:** Peter J. Carrick.

**Data curation:** Katherine J. Forsythe.

**Formal analysis:** Katherine J. Forsythe.

**Investigation:** Peter J. Carrick, Katherine J. Forsythe.

**Methodology:** Peter J. Carrick, Katherine J. Forsythe.

**Project administration:** Peter J. Carrick.

**Supervision:** Peter J. Carrick.

**Visualization:** Katherine J. Forsythe.

**Writing – original draft:** Peter J. Carrick, Katherine J. Forsythe.

**Writing – review & editing:** Peter J. Carrick.

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
