## [Decision Letter · Decision Letter 0]

14 Feb 2020

PONE-D-19-31206

The species composition - ecosystem function relationship: a global meta-analysis

using data from intact and recovering ecosystems

PLOS ONE

Dear Dr. Carrick,

Thank you for submitting your manuscript to PLOS ONE. After careful consideration, we feel that it has merit but does not fully meet PLOS ONE’s publication criteria as it currently stands. Therefore, we invite you to submit a revised version of the manuscript that addresses the points raised during the review process.

We would appreciate receiving your revised manuscript by Mar 30 2020 11:59PM. To enhance the reproducibility of your results, we recommend that if applicable you deposit your laboratory protocols in protocols.io, where a protocol can be assigned its own identifier (DOI) such that it can be cited independently in the future. For instructions see: http://journals.plos.org/plosone/s/submission-guidelines#loc-laboratory-protocols

We look forward to receiving your revised manuscript.

Kind regards,

Patrice Savadogo, PhD

Academic Editor

PLOS ONE

Additional Editor Comments (if provided):

General comments

Review of PONE-D-19-31206: The species composition - ecosystem function relationship: a global meta-analysis using data from intact and recovering ecosystems

This manuscript meta-analytically reviews literature on biodiversity and ecosystem function (BEF) that involved the outcomes of 25 individual restoration studies. Focusing on the methodological features of the study, the authors appropriately follow the PRISMA guidelines in reporting the findings of their meta analysis and the methods of the literature search and statistical analyses generally appear to be appropriate. There were some areas, however, where it would be beneficial to be more explicit about various methodological details of the meta-analysis. Specifically: (i) The eligibility criteria for studies being chosen for the review: the authors have considered a wide-variety of degradation types in their paper selection that is a great benefit but it is a bit questionable that 4 out of 25 studies were related to insects. Also a strong effort to find all the studies from major biomes would make the study a valuable resource for future studies in this field. One of the reviewer has pointed out that “tropical systems have severely weak represented”. (ii) The author should clarify whether all the studies included in the pooled meta-analysis have a comparable diagnostic cut-off point.

Based on the above and from the referees' comments, this paper require is a Major revisions. Please use the comments to make a thorough and careful revision.

Journal Requirements:

Reviewers' comments:

Reviewer's Responses to Questions

**Comments to the Author**

1. Is the manuscript technically sound, and do the data support the conclusions?

Reviewer #1: Partly

Reviewer #2: Yes

2. Has the statistical analysis been performed appropriately and rigorously? 

Reviewer #1: No

Reviewer #2: Yes

3. Have the authors made all data underlying the findings in their manuscript fully available?

Reviewer #1: Yes

Reviewer #2: No

4. Is the manuscript presented in an intelligible fashion and written in standard English?

Reviewer #1: Yes

Reviewer #2: Yes

5. Review Comments to the Author

Reviewer #1: The authors have conducted a meta-analysis of the relationship between species diversity (via the Bray-Curtis index) and a variety of ecosystem function metrics. The topic is timely. A willingness to re-examine long-held relationships in the field is essential to continued growth, and this paper could make a valuable contribution in that sense. However, I have concerns that the analysis, as presented, lacks sufficient documentation and breadth to support the authors’ conclusions. My largest issues are:

1. Representation of study scope. The study is repeatedly presented as a wide-reaching and “global” meta-analysis, but only 25 studies were analyzed, 21 of those were plant studies (the other four were insects), and tropical systems have severely weak represented. The authors should make such limitations in taxa, region, and study size more explicit in scope of their paper and conclusions.

2. Lack of documentation on study comparability. One of the greatest challenges of a meta analysis is ensuring studies are suitably comparable in their response metric, as well as the attributes of the study systems themselves. Currently, it is difficult to evaluate how the authors addressed this challenge. Though the authors have created a standardized summary statistic, this does not address underlying qualitative differences (e.g. species, region, study methods) that might create unaccounted variation (i.e. the apples to oranges problem). This unaccounted variation could contribute to the weak correlation reported by the authors. Additionally, the authors’ summary statistic is a simple mean, and so does not account for within-study variance in the response variable. Koricheva et al.’s Handbook of meta-analysis in ecology and evolution is a useful resource for addressing these issues.

3. Focus on a single similarity metric. A central premise of the paper is that species composition has greater impact than species richness. Why not, then, calculate both composition and richness, instead of composition alone? To me, the most powerful approach would be to compare these two measures within a common system to see if/how the results truly differ. Analyzing results from multiple metrics would make the conclusions more robust and would make better use of the power of meta analysis.

L77-78: I would appreciate more detail on what specific components/definitions of ecosystem functioning (e.g. nutrient cycling, primary productivity, food production, etc.) the authors are referring to here and in other parts of the introduction. This would help define the scope of the paper and be more illustrative to readers not as familiar with the field.

L80-81: What breadth of regions/taxa do these studies represent?

L108: How are “intact natural ecosystems” defined? Are the systems under consideration similar in region/biome/size?

L119-120: While I agree it is important to consider how ecosystems respond to real-world change driven by factors such as “degradation, restoration, fragmentation or climate change,” these processes likely have very different trajectories/outcomes in terms of species composition and ecosystem function. The authors might consider clarifying this, as well as what scope of altered landscapes are under consideration in this paper.

L140-143: On what evidence is this claim based? How far is “sufficiently far”?

L171-172: Some info on studies is available in Table 2, but more summary information would be helpful. How many studies were available in each habitat category? What species/regions are represented, and what are their distributions? What types of restoration projects do the studies represent? What spatial/temporal scales do the studies cover? One of the most challenging aspects of a meta analysis is ensuring the selected studies are sufficiently comparable, so providing more summary information on the studies is essential.

L172: What is the difference between forest and woodland?

L196: What are some examples of an “intact ecosystem” across the different studies considered? Are they comparable?

L214: How does this equation incorporate variance in the measurements? As far as I can see, only the mean values are being used. A more informative metric might consider how much the two distributions overlap, not just how far the two means are. Otherwise, you could have two mean values that are far apart, but only because the two distributions have wide variance.

L222-224: How similar were measures of ecosystem function across studies? (in terms of the specific metrics used, not the categories the authors assigned). This information is important to evaluate whether the measurements are truly comparable, and even when standardizing them into ratios.

L233: I am not sure that a linear model is the most appropriate way to analyze these data. My concern is that the explanatory variables, ecosystem function and species composition, are not wholly independent — both measurements are calculated for each group of reference sites. I imagine there is large variation among different groups of reference sites, and I don’t see how the model is currently accounting for this. Would it not be better to use something like a paired T-test, where the two measures are directly compared within the context of their site?

L243-246: The authors should consider including additional variables in the model that might explain variation among sites, such as study species, region, study size, etc. These factors could create variation among studies that will mask overall trends.

Table 2: Looking at the table, I am concerned about the representativeness and comparability of the data. First, the studies included in analysis are primarily plants. I believe the authors should make this clearer upfront, and structure the scope of their questions/conclusions accordingly. Second, how comparable are ants to vegetation? And what kinds of vegetation — grasses, trees? Finally, the study is described multiple places as “global,” but this is a bit of a micharacterization, as there are 10 studies from Australia, 7 from North America, 5 from Europe, 2 from Asia, 1 from South America, and 0 from Africa. This regional bias toward temperate systems should be at the very least addressed, preferably incorporated more directly in analysis, especially given the high richness of species and ecosystem services in tropical areas that is currently not being represented.

L316: I believe the reference figure should be Fig 3, not Fig 2.

Table 3: Authors should report a full list of models run (not just the best-fitting model), as well as the AICc values. This would be fine as a supplementary table.

Fig 4/5: Would it be possible to report both raw data and the modeled response in the same figure? This would make interpretation easier and take up less figure space. Similar thoughts for fig 6/7.

L426-429: For a relationship like this, I would expect time to be an important factor. Among the studies included, what amount of time had the sites been degraded? How long had restoration been taking place? Can these factors be accounted for in the model?

L48: The authors use the Bray-Curtis index and find limited evidence for a relationship with ecosystem functioning. Would this result hold true with other composition/richness indices? How does choice of measurement influence results? Conclusions would be more robust if multiple indices were considered.

L441: But this study did not analyze species richness, despite the seeming ease of doing so. Why not report results from both Bray-Curtis and richness, so as to compare them directly within a common analytical framework? This would make for a more robust result and take better advantage of the meta analysis capabilities.

L507-509: What sizes and separation distances are represented by these sites? How many of the cited examples are plants? Are there any contrary examples that might be illustrative? I ask because (1) I would expect studies with smaller plots or greater distance between plots to have higher species turnover, and (2) I would expect plants to have higher turnover due to localized growth patterns of some species, as compared to more mobile vertebrate taxa.

L521-525: What range of species richnesses is represented by the studies? I do not recall seeing this reported.

Reviewer #2: Comments to the Corresponding author

The manuscript “The species composition - ecosystem function relationship: a global meta-analysis using data from intact and recovering ecosystems ” presents a well written study on the relationship between species composition and ecosystem function. The authors used formal meta-analysis to synthesize the outcomes of 25 individual restoration studies. At the surface, this study is about the species composition - ecosystem function relationship, but deep down, it asks a more fundamental question: Is there scientific evidence that the results from constructed BEF experiments can be transferred to real world ecosystems - as is often implied by BEF studies? Thereby, the manuscript addresses an long-standing topic and open question in ecology which makes it an immensely valuable contribution to the scientific literature around BEF studies. The manuscript underpins these questions with a sound methodological synthesis approach which is the true strength of formal meta-analyses.

The methodology is sound, the analysis is sophisticated and the conclusions drawn from the results of the study are valid. However, the results are not clearly presented and the discussion section requires much more focus. I’ll try to list some points for improvement below:

Abstract

- Overall, I find the abstract to be the weakest part of the manuscript, but it can be improved easily. For example, as someone new to the topic of restoration research I was confused how “reference sites in restoration studies” (l50) differ from “degraded but restored sites” (l55)? After going through the manuscript I think I now know what is meant here, but since PLOS One caters to a very wide and interdisciplinary audience the abstract should be more easy to access. Please rephrase accordingly.

- l56-l57 Please rephrase

- l58 The six types of EF mentioned here come somewhat out of the blue. Either mention which these six are or remove.

- l62-65 I don’t see how these conclusions can been drawn from what has been written so far in the abstract. Please change to a concluding sentence that allows the abstract to stand on its own.

- Please mention how many studies and measures were used in the meta analyses in the abstract

Introduction

- l75ff When reflecting on the BEF field, consider swapping out the term biodiversity for species richness. Many of the the cited studies actually focus on SR.

- the literature cited in the entire paper seems a bit outdated with the most recent reference being from 2017. There have been many, also topically suitable studies been published in the last couple of years. Please bring the references up to date. Here are some recent meta-analyses which could fit in the manuscript, but there are many, many more available:

Beckmann, M., Gerstner, K., Akin‐Fajiye, M., Ceaușu, S., Kambach, S., Kinlock, N. L., ... & Newbold, T. (2019). Conventional land‐use intensification reduces species richness and increases production: A global meta‐analysis. Global change biology, 25(6), 1941-1956.

Forbes, E. S., Cushman, J. H., Burkepile, D. E., Young, T. P., Klope, M., & Young, H. S. (2019). Synthesizing the effects of large, wild herbivore exclusion on ecosystem function. Functional Ecology, 33(9), 1597-1610.

Taylor, N., Grillas, P., Fennessy, M., Goodyer, E., Graham, L., Karofeld, E., ... & Ross, S. (2019). A synthesis of evidence for the effects of interventions to conserve peatland vegetation: overview and critical discussion. Mires and Peat, 24.

Methods:

- The first section on the “literature search” reads a bit like a textbook description of how one would conduct any meta analysis and does refer very little to the study presented here. I suggest to either move Figure 1 (PRISMA diagram) here and refer to it in the text or put some numbers in the text and move the PRISMA diagram to the appendix (I would prefer the second option). Let the reader know e.g. how many abstracts were screened, how many authors were contacted and how many studies were coded in the end, the text lacks connection to the study otherwise. I am also sure (judging from the numbers in Figure 1) you were not at able to code most studies directly from published sources – a common problem encountered when conducting meta analyses. Consider reflecting on this issue somewhere in your manuscript, either here in the methods or in the discussion section. These sources might be of help:

Gerstner, K., Moreno‐Mateos, D., Gurevitch, J., Beckmann, M., Kambach, S., Jones, H. P., & Seppelt, R. (2017). Will your paper be used in a meta‐analysis? Make the reach of your research broader and longer lasting. Methods in Ecology and Evolution, 8(6), 777-784.

Spake, R., & Doncaster, C. P. (2017). Use of meta-analysis in forest biodiversity research: key challenges and considerations. Forest Ecology and Management, 400, 429-437.

Andivia, E., Villar-Salvador, P., Oliet, J. A., Puértolas, J., & Dumroese, R. K. (2019). How can my research paper be useful for future meta-analyses on forest restoration plantations?. New Forests, 50(2), 255-266.

- PLOS One requires authors to make all data underlying the findings described in their manuscript fully available without restriction. I assume S1 will include that data but I do not have access to it and I wonder if it includes the raw data or the summaries (effect sizes). In case the original authors were asked if they approve publication of the raw data I suggest to do exactly that. If the original authors were not asked, please publish at least the summarized data (is this S1?). Also, please publish the R-code used for the analysis (e.g. on gitHub). Only this way your study becomes fully reproducible. Refer to PLOS One author guidelines which provide further information on where to deposit data and code, the supplementary material might not be the best location.

- The Data Analysis section could be improved by summarizing the performed models in a table and reducing the text accordingly.

Results

- The Literature Search section should be merged with the Methods section, see comments above.

- While informative, Table 2 is also quite wasteful regarding space and it is not very readable. Try reducing the number of rows used by introducing abbreviations and reducing repetition (e.g. only 4 out of 25 studies use something else than vegetation abundance as a species composition measure). Also, please include here the number of species composition measures and EF similarity measures per study.

- L298-306 should be integrated in the methods section as well

- General comment regarding all figures: they are currently of terrible quality and very difficult to read. In fact, Figs 2 and 3 are almost completely black and indecipherable. This might be caused by the submission system but should be improved for re-submission.

- Caption of Figure 4: include the number of studies (25) as well. This way the reader gets the full context. As the text is now it suggests a much larger scope.

- Table 3: Please include results from all tested models not just the base and the best fitting one.

Discussion

- Overall, with 9 pages, the discussion is rather long and should be reduced by at least two and a half pages. I also suggest a restructuring of the discussion to help condensing this section: move subsection “Nature of species composition and other biodiversity relationships” to beginning. Get rid of the contrasting list and shorten the remaining subsections by half each. See below for details.

- The contrasting list is a nice idea but does not really work. Rather write where you confirm the outcomes from expected from BEF studies and where the real world differs from BEF studies (which is actually done a bit later and would be the start of the discussion if you follow my suggestion above).

- Tighten text in l487-509

- l479 - 485 I am not sure what the point is here, one could argue that the 25 studies is a low number itself. I suggest to remove this part.

- l565-575 this largely repeats what has been already said in introduction and earlier in discussion

- l611-613 this does not add much and could be removed

- section in l592-630 can be shortend by half

6. PLOS authors have the option to publish the peer review history of their article (what does this mean?). If published, this will include your full peer review and any attached files.

Reviewer #1: No

Reviewer #2: No

---

## [Author Response · Author response to Decision Letter 0]

24 May 2020

As requested by the editor I have drafter a rebuttal letter that responds to each point raised by the academic editor and each reviewer. This letter was uploaded as separate file and labeled 'Response to Reviewers'. All the editor's and reviewer's comments have been comprehensively addressed as you will see from the attached revised manuscript, figures and supporting information.

The Response to Reviewers rebuttal letter is also comprehensive and the nature of our revision of the manuscript is set out for every point. It is also lengthy and so have not copied the entire document into this space. It is far more readable as the uploaded Word document.

---

## [Decision Letter · Decision Letter 1]

10 Jul 2020

The species composition - ecosystem function relationship: a global meta-analysis using data from intact and recovering ecosystems

PONE-D-19-31206R1

Dear Dr. Carrick,

We’re pleased to inform you that your manuscript has been judged scientifically suitable for publication and will be formally accepted for publication once it meets all outstanding technical requirements.

Kind regards,

Patrice Savadogo, PhD

Academic Editor

PLOS ONE

Additional Editor Comments (optional):

Dear Authors

Your revised manuscript, PONE-D-19-31206R1 has been subjected to a double-blind 2nd-round review process conducted by one of the original referees who reviewed your earlier manuscript and a new one. They both came to the conclusion that you have addressed all comments and suggestions and improved the manuscript accordingly. I agree with their assessment. The manuscript is a good contribution and deserves being published in its present form.

Patrice Savadogo, PhD (Academic Editor)

Reviewers' comments:

Reviewer's Responses to Questions

**Comments to the Author**

1. If the authors have adequately addressed your comments raised in a previous round of review and you feel that this manuscript is now acceptable for publication, you may indicate that here to bypass the “Comments to the Author” section, enter your conflict of interest statement in the “Confidential to Editor” section, and submit your "Accept" recommendation.

Reviewer #2: All comments have been addressed

Reviewer #3: All comments have been addressed

2. Is the manuscript technically sound, and do the data support the conclusions?

Reviewer #2: Yes

Reviewer #3: Yes

3. Has the statistical analysis been performed appropriately and rigorously? 

Reviewer #2: Yes

Reviewer #3: Yes

4. Have the authors made all data underlying the findings in their manuscript fully available?

Reviewer #2: Yes

Reviewer #3: Yes

5. Is the manuscript presented in an intelligible fashion and written in standard English?

Reviewer #2: Yes

Reviewer #3: Yes

6. Review Comments to the Author

Reviewer #2: The authors have addressed all of my comments in a satisfactory manner and I therefore recommend acceptance of the manuscript.

Reviewer #3: Authors have addressed most of reviewer's suggestions and improved the manuscript accordingly. When authors did not agree with comments of reviewer 1, they have explained their reasons convincingly. The manuscript is a good contribution and deserves being published in its present form.

7. PLOS authors have the option to publish the peer review history of their article (what does this mean?). If published, this will include your full peer review and any attached files.

Reviewer #2: No

Reviewer #3: No

---

## [Editor Report · Acceptance letter]

17 Jul 2020

PONE-D-19-31206R1 

The species composition - ecosystem function relationship: a global meta-analysis using data from intact and recovering ecosystems 

Dear Dr. Carrick:

I'm pleased to inform you that your manuscript has been deemed suitable for publication in PLOS ONE. Congratulations! Your manuscript is now with our production department. 

Kind regards, 

on behalf of

Dr. Patrice Savadogo 

Academic Editor

PLOS ONE